# Knowledge Intensive Learning of Credal Networks

**Saurabh Mathur**[1]  **Alessandro Antonucci**[2]  **Sriraam Natarajan**[1]

[1]Erik Jonsson School of Engineering & Computer Science, University of Texas at Dallas, Richardson, Texas, USA
[2]Istituto Dalle Molle di Studi sull'Intelligenza Artificiale (IDSIA) - Lugano, Switzerland

## Abstract

Bayesian networks are a popular class of directed probabilistic graphical models that allow for closed-form learning of the local parameters if complete data are available. However, learning the parameters is challenging when the data are sparse, incomplete, and uncertain. In this work, we present an approach to this problem based on *credal networks*, a generalization of Bayesian networks based on set-valued local parameters. We derive an algorithm to learn such set-valued parameters from data using qualitative knowledge in the form of monotonic influence statements. Our empirical evaluation shows that using qualitative knowledge reduces uncertainty about the parameters without significant loss in accuracy.

## 1 INTRODUCTION

Bayesian networks (BNs) are a powerful tool for representing and reasoning under uncertainty. They have been successfully applied in a wide variety of domains [Daly et al., 2011] including healthcare [Lucas et al., 2004], weather forecasting [Abramson et al., 1996], software engineering [Pendharkar et al., 2005] and risk management [Fan and Yu, 2004]. However, BNs require complete and accurate data to learn the network parameters from data. In many real-world applications, such data may not be fully available.

To overcome the limitations of noisy and sparse data, domain knowledge might be used to learn BNs. Domain knowledge can concisely determine the direction and strength of relationships between variables [Niculescu et al., 2006] and trends in these relationships [Wellman, 1990]. Incorporating domain knowledge has been studied more broadly in machine learning. Knowledge in the form of precision-recall trade-off [Yang et al., 2014], label preferences [Odom et al., 2015], privileged information and qualitative influence statements [Altendorf et al., 2005, Yang and Natarajan, 2013, Mathur et al., 2023b,a] have been successfully used to learn more accurate and robust models. While these methods overcome the limitations of noisy and sparse data, they can still not deal with incomplete and uncertain data.

Credal networks (CNs) address such a limitation by extending BNs to explicitly represent incompleteness and uncertainty about probability distributions [Mauá and Cozman, 2020]. They provide a more cautious approach to the specification of probabilistic models. This makes CNs especially useful for noisy, sparse, and incomplete data domains. However, inducing them purely from data can make the model too "imprecise" and result in vacuous inferences.

Inspired by knowledge-guided learning of probabilistic models, we present a solution to the problem of learning accurate yet robust models in the presence of noisy, sparse, and possibly incomplete data by embedding domain knowledge in CNs. Specifically, we consider a subclass of qualitative influence statements called *monotonic influence statements* to make CNs more precise. The main contributions of this paper are the development of a learning method for CNs that effectively exploits monotonic influence relationships in the domain as knowledge and the preliminary empirical evaluation of the corresponding learning algorithm.

Specifically, we make the following key contributions: (i) we propose the first method for learning CNs from data and domain knowledge; (ii) we consider a specific type of knowledge – monotonic influences as qualitative constraints – to learn set-valued parameters; (iii) we demonstrate the effectiveness and efficacy of the learning algorithm on a combination of benchmark, BNs, based on three real healthcare data sets and a high-impact real-world problem of mitigating adverse pregnancy outcomes.

The rest of this paper is organized as follows: after providing background about CNs and qualitative influences, we present our method for learning CNs from data using domain knowledge. We then present our empirical evaluation and conclude with a discussion of central outlooks.

*Accepted for the 40th Conference on Uncertainty in Artificial Intelligence* (UAI 2024).

## 2 BACKGROUND CONCEPTS

**Bayesian and credal networks.** *Bayesian networks* (BNs, Koller and Friedman 2009) are probabilistic graphical models that compactly represent joint *probability mass functions* (PMFs). Formally, a BN over a set of variables $\boldsymbol{X} = \{X_1, \ldots, X_n\}$ is a pair $\langle \mathcal{G}, \theta \rangle$. Here, $\mathcal{G}$ is a directed acyclic graph such that each node corresponds to a random variable in $\boldsymbol{X}$ and $\theta$ is a set of conditional PMFs specified for each variable, given all the possible values of its parents $\mathrm{Pa}_X \subset \boldsymbol{X}$ according to $\mathcal{G}$. Graph $\mathcal{G}$ represents conditional independence relations according to the Markov condition. As a result, the joint PMF induced by the BN can be expressed as the following factorization:

$$P(\boldsymbol{x}) = \prod_{X \in \boldsymbol{X}} P(x|\mathrm{pa}_X), \qquad (1)$$

for each state $\boldsymbol{x} \in \mathrm{Dom}(\boldsymbol{X})$, where $\mathrm{pa}_X \in \mathrm{Dom}(\mathrm{Pa}_X)$ and $x \in \mathrm{Dom}(X)$ are the states consistent with $\boldsymbol{x}$.

*Credal networks* (CNs, Mauá and Cozman 2020) are a generalization of BNs that allows us to define sets of joint PMFs. A set of PMFs over $X$ is called *credal set* (CS) and denoted as $K(X)$. CSs [Levi, 1980, Augustin et al., 2014] allow us to explicitly represent incompleteness in uncertain specifications (e.g., a *vacuous* CS including all the possible PMFs over $X$, thus expressing a condition of complete ignorance). In this work, we consider closed and convex CSs, that are also finitely-generated, i.e., induced by the convex closure of a finite number of linear constraints on the PMFs $P(X)$ belonging to $K(X)$. This allows us to equivalently describe each conditional CS by listing its extreme points, whose number should be also finite.

In practice, the specification of a CN is the same as that of a BN except that each (conditional) PMF is replaced by a CS. The Markov condition can also be applied to CNs, provided that a suitable notion of independence is considered. Here we focus on the notion of *strong* independence, i.e., $X$ and $X'$ are independent according to CS $K(X, X')$ if they are independent in the stochastic sense for each PMF in the extreme points of the CS. This allows us to define a joint CS $K(\boldsymbol{X})$ as the convex closure of the set of all joint PMFs as in Eq. (1) such that the conditional PMFs are taken from the conditional CSs in the CN specification or, equivalently, from their vertices [Antonucci and Zaffalon, 2008]. Inferences in CNs are consequently intended as the computation of the lower and upper bounds of a BN query w.r.t. such a joint CS. In spite of the hardness of the general inference [Mauá et al., 2014], exact [Cabañas and Antonucci, 2021] and approximate [Antonucci et al., 2015] schemes to query possibly large CNs are available.

**Decision-making in CSs.** Recall that decision-making in PMFs involves finding the state (decision) that minimizes a given loss function. With 0-1 losses, this corresponds to

taking as optimal state $x^* := \arg\max_{x \in \boldsymbol{X}} P(x)$. Decision-making in CSs can be done using *interval dominance* [Zaffalon, 2002, Troffaes, 2007]. State $x \in \mathrm{Dom}(X)$ is said to interval-dominate another state $x' \in \mathrm{Dom}(X)$ according to the CS $K(X)$ if and only if:

$$\min_{P(X) \in K(X)} P(x) > \max_{P(X) \in K(X)} P(x'), \qquad (2)$$

where the two optimizations can be computed w.r.t. the linear constraints in the CS specification, or, equivalently, by only considering the extreme points. If a single state interval-dominates all other states, then that state can be selected as optimal for the decision. However, we might have more than one undominated state. In such cases, we can abstain from making a further decision and regard all the undominated states as optimal.

**Learning CSs.** The *imprecise Dirichlet model* (IDM, Walley 1996) is the most popular approach for learning CSs from categorical data. This is a generalization of a Bayesian approach combining a multinomial likelihood with a Dirichlet prior distribution. Instead of a single Dirichlet prior, the IDM posits a set of priors, called the imprecise Dirichlet prior, including all the Dirichlet prior distributions of given *equivalent sample size* (ESS). Specifically, when learning from a data set $\mathcal{D}$ of observations of the random variable $X$, the set of Dirichlet priors is parameterized as $\mathrm{Dir}(st_X)$. Here, $s \in \mathbb{R}^+$ is the ESS and $t_X := \{t_x\}_{x \in \mathrm{Dom}(X)}$ with $t_x \in [0, 1]$ and $\sum_x t_x = 1$. The probability induced by the IDM is therefore:

$$P(x) = \frac{N_x + st_x}{N + s}, \qquad (3)$$

where $N_x$ is the number of times $X = x$ occurs in data and $N$ is the total number of observations in the $\mathcal{D}$. The bounds w.r.t. the imprecise Dirichlet prior are therefore:

$$\underline{P}(x) := \min_{P(X) \in K(X)} P(x) = \min_{t_x \in [0,1]} \frac{N_x + st_x}{N + s} = \frac{N_x}{N + s}, \quad (4)$$

$$\overline{P}(x) := \max_{P(X) \in K(X)} P(x) = \max_{t_x \in [0,1]} \frac{N_x + st_x}{N + s} = \frac{N_x + s}{N + s}, \quad (5)$$

for each $x \in \mathrm{Dom}(X)$. Those bounds induce linear constraints on a PMF $P(X)$, thus defining a CS $K(X)$. Note that, for data sets whose cardinality is small w.r.t. the ESS $s$, these bounds can be quite broad. In the rest of the paper, we discuss a procedure based on domain knowledge to shrink these bounds.

**Domain knowledge as qualitative influence statements.** Qualitative influence statements (QISs, Wellman 1990) describe the influence of one or more variables over another variable. They allow domain experts to concisely express a trend in the distribution without needing to specify precise values. Here we focus on learning CNs using a class of QISs called *monotonic influence statements* (MISs, Altendorf et al. 2005). MISs refer to ordinal, and hence also

Boolean as a special case, variables. Given a variable $Y$ and a joint variable $\boldsymbol{X}$ in a probabilistic model, we say that $Y$ is *positively monotonically influenced* by parent $X \in \boldsymbol{X}$ if higher values of $X$ stochastically result in higher values of $Y$, *ceteris paribus* (i.e, the value of all other parents held constant). Such an influence is denoted as $X \overset{M+}{\prec} Y$ and corresponds to domain knowledge of the form "as $X$ increases, $Y$ also increases". We express such a MIS as the inequality:

$$P(Y \le y | x, \tilde{\boldsymbol{x}}) \ge P(Y \le y | x', \tilde{\boldsymbol{x}}) \qquad (6)$$

for each $x, x' \in \mathrm{Dom}(X)$ such that $x \le x'$, $y \in \mathrm{Dom}(Y)$, and $\tilde{\boldsymbol{x}} \in \mathrm{Dom}(\tilde{\boldsymbol{X}})$, where $\tilde{\boldsymbol{X}} := \boldsymbol{X} \setminus \{X\}$. Negative influence can be defined analogously and denoted as $X \overset{M-}{\prec} Y$.

**Related work.** QISs have been used to induce more accurate precise probabilistic models from noisy and sparse data for both discriminative [Kokel et al., 2020, Odom et al., 2015] and generative learning settings [van der Gaag et al., 2004, Altendorf et al., 2005, de Campos et al., 2008, Yang and Natarajan, 2013, Plajner and Vomlel, 2020, Mathur et al., 2023b,a]. In this work, we deal with learning imprecise generative models from sparse, incomplete, and uncertain data. QISs have been previously used to make generative models more precise. Renooij and van der Gaag [2002] introduce influence-intervals and perform interval-propagation on qualitative probabilistic networks to shrink the intervals. In contrast, our method maintains probabilistic semantics by dealing with (closed and convex) CSs. QISs have also been used to learn conditional CSs. de Campos and Cozman [2005] use qualitative influences as constraints on the imprecise Dirichlet prior distributions. However, in the presence of prior-data conflicts [Evans and Moshonov, 2006], this approach does not guarantee consistency with the qualitative knowledge.

Our approach of directly constraining a CS provides a more flexible solution to this problem. This also makes it independent of the way that the CS is initially computed.

## 3 KNOWLEDGE-INTENSIVE LEARNING

We aim to improve the performance of CN models by incorporating qualitative domain knowledge into the learning process. The key idea is that qualitative knowledge can serve as a strong inductive bias. While one could envision sampling data from QISs/MISs (as a single piece of knowledge could generalize several data points in one fell swoop), we take a different approach of using the knowledge to define constraints on the learning model.

From a Bayesian perspective, qualitative knowledge might guide the specification of the prior distribution (e.g., a comparative judgment inducing an analogous constraint on the corresponding parameters of a Dirichlet distribution). Yet, in the presence of prior-data conflicts [Evans and Moshonov,

2006], the Bayesian approach does not guarantee consistency with the qualitative knowledge. CSs approaches are known to provide a more flexible solution to this problem [Walter and Augustin, 2009]. The problem of integrating qualitative knowledge (and in particular MISs) in the statistical learning of a credal model corresponds to the following learning task:

---

**Given:** Data set $\mathcal{D} := \{y^{(i)}, \boldsymbol{x}^{(i)}\}_{i=1}^{N}$ over variables $(Y, \boldsymbol{X})$ and a collection $C$ of MISs as in Eq. (6).
**To Do:** Learn a collection of conditional CSs over $Y$, say $\{K(Y|\boldsymbol{x})\}_{\boldsymbol{x} \in \mathrm{Dom}(\boldsymbol{X})}$, that are compatible with $C$.

---

**A healthcare example.** The above learning setting is crucial in several domains, including healthcare, that require cautious models to be learned from limited and noisy data sets. As an example, consider a simplistic problem of modeling *Gestational Diabetes Mellitus* ($G$) based on two risk factors – when the age at the start of pregnancy is greater than 35 ($A$) and when the Body Mass Index at the start of pregnancy greater than 25 ($B$). Data-driven methods like the IDM can be used to learn the *conditional* CSs (CCSs) for $G$ given $A$ and $B$. However, small and noisy datasets can induce wide bounds, making predictions uninformative.

Our intuition, to be empirically tested, is that domain knowledge, specifically in the form of qualitative constraints could significantly shorten the (credal) bounds, thus leading to an actionable outcome. In our example, we might know that both age at the start of pregnancy and the Body Mass Index positively monotonically influence the risk of Gestational Diabetes Mellitus. This knowledge can then be used to filter out the PMFs that violate this rule to obtain narrower and more informative bounds.

Table 1 illustrates this approach. The columns on the left show the CCSs learned purely from data, while those on the right presents the CCSs obtained after filtering PMFs that are not compatible with the domain knowledge. While none of the intervals for $G$ dominate for any combination of $A$ and $B$ for the CCSs on the left, the CCSs on the right has a configuration ($\{A = 1, B = 1\}$) where the interval corresponding to $G = 1$ interval-dominates the one corresponding to $G = 0$.

### 3.1 OUR APPROACH

We approach the above problem by obtaining an initial set of CCSs from the data set $\mathcal{D}$ (for eg, by standard IDM learning) and then shrinking the bounds by eliminating the PMFs that violate the qualitative influence constraints (MISs) in $C$.

Let the initial set of CCSs be $\underline{P}_0(y|\boldsymbol{x})$ and $\overline{P}_0(y|\boldsymbol{x}), \forall \boldsymbol{x} \in \mathrm{Dom}(\boldsymbol{X}), y \in \mathrm{Dom}(Y)$. Without loss of generality, we obtain new upper bounds by finding the largest values

$$\overline{P}(y|\boldsymbol{x}) \forall y \in \mathrm{Dom}(Y), \boldsymbol{x} \in \mathrm{Dom}(\boldsymbol{X}) \qquad (7)$$

| $A$ | $B$ | $P(G=0 \mid A,B)$ | $P(G=1 \mid A,B)$ | $A$ | $B$ | $P(G=0 \mid A,B)$ | $P(G=1 \mid A,B)$ |
|---|---|---|---|---|---|---|---|
| 0 | 0 | $[0.4, 0.7]$ | $[0.3, 0.6]$ | 0 | 0 | $[0.5, 0.7]$ | $[0.3, 0.5]$ |
| 0 | 1 | $[0.3, 0.7]$ | $[0.3, 0.7]$ | 0 | 1 | $[0.3, 0.5]$ | $[0.5, 0.7]$ |
| 1 | 0 | $[0.0, 1.0]$ | $[0.0, 1.0]$ | 1 | 0 | $[0.4, 0.5]$ | $[0.5, 0.6]$ |
| 1 | 1 | $[0.0, 0.7]$ | $[0.3, 1.0]$ | 1 | 1 | $[0.0, 0.2]$ | $[0.8, 1.0]$ |

Table 1: CCSs for the presence of Gestational Diabetes ($G$) given two risk factors – age at pregnancy greater than 35 ($A$) and BMI greater than 25 ($B$) estimated from a small sample of data using the IDM (left) and the CCS obtained by eliminating the PMFs that were incompatible with the knowledge that both risk factors positively monotonically influence the risk of G ($A_{\prec}^{M+}G$ and $B_{\prec}^{M+}G$ respectively).

that satisfy the monotonicity constraints. This is equivalent to the following constrained optimization problem:

$$
\underset{\substack{\underline{P}_0(y|\boldsymbol{x}) \leq q_{y|\boldsymbol{x}} \leq \overline{P}_0(y|\boldsymbol{x}) \\ q_{y|\boldsymbol{x}} \models C \\ \boldsymbol{x} \in \mathrm{Dom}(\boldsymbol{X}) \\ y \in \mathrm{Dom}(Y)}}{\arg\max} \mathcal{L}(\boldsymbol{q}), \tag{8}
$$

where $\boldsymbol{q} := \{q_{y|\boldsymbol{x}}\}_{\boldsymbol{x} \in \mathrm{Dom}(\boldsymbol{X})}^{y \in \mathrm{Dom}(Y)}$ is the set of all optimization variables, the objective function $\mathcal{L}(\boldsymbol{q})$ is defined as

$$
\mathcal{L}(\boldsymbol{q}) := \sum_{\substack{\boldsymbol{x} \in \mathrm{Dom}(\boldsymbol{X}) \\ y \in \mathrm{Dom}(Y)}} q_{y|\boldsymbol{x}}, \tag{9}
$$

and $q_{y|\boldsymbol{x}} \models C$ denotes that the optimization variables entail the MIS constraints in $C$ as stated by Eq. (6). Note that $C$ imposes constraints across the CCSs corresponding to different configurations of the parents. So, if $C = \emptyset$, then Eq. (8) becomes equivalent to performing separate optimizations for each $q_{y|\boldsymbol{x}}$ which recover the initial CCSs. An analogous optimization can be considered for the lower bounds.

However, such linear programs are not guaranteed to have feasible solutions because some constraints might be unsatisfiable under the initial bound constraints. If this is the case we address the optimization using the *barrier penalty* method [Luenberger and Ye, 2016]. Specifically, we encode each MIS constraints $c \in C$ of the form $X_{\prec}^{M+}Y$ as $\delta_c(\boldsymbol{q}, \epsilon) \leq 0$ where:

$$
\delta_c(\boldsymbol{q}, \epsilon) = \sum_{y' \leq y} q_{y'|x', \tilde{\boldsymbol{x}}} - \sum_{y'' \leq y} q_{y''|x, \tilde{\boldsymbol{x}}} + \epsilon, \tag{10}
$$

and we introduce a penalty, $\max\{0, \delta_c(\boldsymbol{q}, \epsilon)\}^2$. Now, instead of Eq. (8), we solve a sequence of optimization problems of the form:

$$
\underset{\substack{\underline{P}_0(y|\boldsymbol{x}) \leq q_{y|\boldsymbol{x}} \leq \overline{P}_0(y|\boldsymbol{x}) \\ \boldsymbol{x} \in \mathrm{Dom}(\boldsymbol{X}) \\ y \in \mathrm{Dom}(Y)}}{\arg\max} \left[ \mathcal{L}(\boldsymbol{q}) - \lambda \underbrace{\sum_{c \in C} \max\{0, \delta_c(\boldsymbol{q}, \epsilon)\}^2}_{\text{Penalty}} \right], \tag{11}
$$

for $\lambda = 10^0, 10^1, 10^2, \ldots, 10^L$ until the penalty term vanishes, where $\mathcal{L}(\boldsymbol{q})$ is the objective function in Eq. (9). If a feasible solution exists, then this method is guaranteed to converge to a solution in the limit [Luenberger and Ye, 2016]. We analogously proceed for the minimization task.

## 3.2 GRADIENTS

As outlined in Eq. (11), solving a series of optimization problems forms the core of our method. Each such optimization problem can be solved using a standard gradient ascent procedure that supports parameter bounds of the form

$$
\underline{P}_0(y|\boldsymbol{x}) \leq q_{y|\boldsymbol{x}} \leq \overline{P}_0(y|\boldsymbol{x}) \ \forall \boldsymbol{x} \in \mathrm{Dom}(\boldsymbol{X}), \\ y \in \mathrm{Dom}(Y) \tag{12}
$$

We now present the details of the gradients of the objective function with respect to each element $q_{y_i|\boldsymbol{x}_j}$ of the parameter vector $\boldsymbol{q}$. The gradient of the objective function in Eq. (11) with respect to each $q_{y_i|\boldsymbol{x}_j}$ of $\boldsymbol{q}$ is

$$
\frac{\partial}{\partial q_{y_i|\boldsymbol{x}_j}} [\mathcal{L}(\boldsymbol{q}) - \lambda \sum_{c \in C} \max\{0, \delta_c(\boldsymbol{q}, \epsilon)\}^2]
$$
$$
= \frac{\partial \mathcal{L}(\boldsymbol{q})}{\partial q_{y_i|\boldsymbol{x}_j}} - \lambda \sum_{c \in C} 2\max\{0, \delta_c(\boldsymbol{q}, \epsilon)\} \frac{\partial \max\{0, \delta_c(\boldsymbol{q}, \epsilon)\}}{\partial q_{y_i|\boldsymbol{x}_j}}
$$
$$
= 1 - \lambda \sum_{c \in C} 2\max\{0, \delta_c(\boldsymbol{q}, \epsilon)\} \mathbb{1}_{\delta_c(\boldsymbol{q}, \epsilon) > 0} \frac{\partial \delta_c(\boldsymbol{q}, \epsilon)}{\partial q_{y_i|\boldsymbol{x}_j}}. \tag{13}
$$

Here, the gradient of the $\delta_c(\boldsymbol{q}, \epsilon)$ term with respect to each $q_{y_i|\boldsymbol{x}_j}$ of $\boldsymbol{q}$ is

$$
\frac{\partial \delta_c(\boldsymbol{q}, \epsilon)}{\partial q_{y_i|\boldsymbol{x}_j}} = \frac{\partial}{\partial q_{y_i|\boldsymbol{x}_j}} \sum_{y' \leq y} q_{y'|x', \tilde{\boldsymbol{x}}} - \sum_{y'' \leq y} q_{y''|x, \tilde{\boldsymbol{x}}} + \epsilon
$$
$$
= \sum_{y' \leq y} \frac{\partial q_{y'|x', \tilde{\boldsymbol{x}}}}{\partial q_{y_i|\boldsymbol{x}_j}} - \sum_{y'' \leq y} \frac{\partial q_{y''|x, \tilde{\boldsymbol{x}}}}{\partial q_{y_i|\boldsymbol{x}_j}} \tag{14}
$$

Once these gradients are obtained, we can solve the optimization problem in Eq. (11) for a given value of $\lambda$ using a gradient ascent procedure with parameter bounds.

**Algorithm 1** ConstrOpt

**Input**:

$\sigma$ (+1 if maximize and -1 if minimize)

$\{\underline{P}(y|\boldsymbol{x}), \overline{P}(y|\boldsymbol{x})\}_{y\in\text{Dom}(Y), \boldsymbol{x}\in\text{Dom}(\boldsymbol{X})}$ (CS bounds)

$C$ (MISs)

$t_{\max}$ (maximum number of iterations)

**Output**:

upper/lower CS bounds satisfying $C$

1: Initialize $\boldsymbol{q} = \underset{\substack{\underline{P}(y|\boldsymbol{x})\leq q_{y|\boldsymbol{x}}\leq\overline{P}(y|\boldsymbol{x}) \\ \boldsymbol{x}\in\text{Dom}(\boldsymbol{X}) \\ y\in\text{Dom}(Y)}}{\arg\max}\ \sigma\mathcal{L}(\boldsymbol{q})$

2: $\lambda = 1, t = 1$

3: **while** $\sum_{c\in C}\max\{0, \delta_c(\boldsymbol{q}, \epsilon)\}^2 > 0$ and $t \leq t_{\max}$ **do**

4: $\quad \boldsymbol{q} = \arg\max\left[\sigma\mathcal{L}(\boldsymbol{q}) - \lambda\sum_{c\in C}\max\{0, \delta_c(\boldsymbol{q}, \epsilon)\}^2\right]$

5: $\quad \lambda = \lambda \times 10$

6: $\quad t = t + 1$

7: **end while**

8: **return** $\boldsymbol{q}$

## 3.3 ALGORITHM

We use these gradients and parameter bounds to optimize the objective function in Eq. (11) using the L-BFGS-B algorithm. We describe the procedure to solve the series of optimization problems in Algorithm 1. To perform the maximization (or minimization) we start with the upper bound (or the lower bound) and solve a series of optimization problems of the form described in Eq. (11). For each of these optimization problems, we use the previous solution as the initialization and we increase the value of the penalty weight $\lambda$ by a factor of 10 to allow for a jump start and early convergence.

Algorithm 2 details our procedure (KnowLearnCCS for Knowledge driven learning of Conditional Credal Sets) to obtain the consistent conditional CSs from the data set $\mathcal{D}$ and the MISs $C$. The algorithm begins by computing the IDM conditional CSs from $\mathcal{D}$. It then uses the MISs $C$ to shrink the CS bounds. It does so by finding the highest and lowest values in the initial CS that satisfy all the constraints in $C$. These values are obtained by constrained optimization based on the barrier penalty method. This is performed by sub-procedure detailed by Algorithm 1.

## 4 EXPERIMENTAL EVALUATION

High-stakes domains like healthcare require models that support cautious decision-making. While data-driven approaches like the IDM can learn CSs induced by upper and lower bounds on the PMFs, these sets can be too broad when learned from small and noisy data sets as is common in such

**Algorithm 2** KnowLearnCCS

**Input**:

$\mathcal{D}$ (data set over $\boldsymbol{X}$ and $Y$)

$C$ (MISs)

$t_{\max}$ (maximum number of iterations)

**Output**:

CS bounds

1: Initialize $\underline{P}(y|\boldsymbol{x}) = \underline{P}_0(y|\boldsymbol{x}), \overline{P}(y|\boldsymbol{x}) = \overline{P}_0(y|\boldsymbol{x})$ for each $y \in \text{Dom}(Y)$ and $\boldsymbol{x} \in \text{Dom}(\boldsymbol{X})$

2: $\{\overline{P}(y|\boldsymbol{x})\}_{y,\boldsymbol{x}} = \text{ConstrOpt}(+1, \underline{P}_0, \overline{P}_0, C, t_{\max})$

3: $\{\underline{P}(y|\boldsymbol{x})\}_{y,\boldsymbol{x}} = \text{ConstrOpt}(-1, \underline{P}_0, \overline{P}_0, C, t_{\max})$

4: **return** $\{[\underline{P}(y|\boldsymbol{x}), \overline{P}(y|\boldsymbol{x})]\}_{y\in\text{Dom}(Y), \boldsymbol{x}\in\text{Dom}(\boldsymbol{X})}$

domains. We hypothesize that qualitative domain knowledge can be used to eliminate inconsistent PMFs from such CSs making them more informative for decision-making while remaining cautious. Concretely, we aim to answer the following research questions:

**(Q1)** Can MISs be used to improve the coverage of a collection of CCSs in small and noisy data sets?

**(Q2)** Does imposing MIS constraints directly on the posterior distribution result in more accurate models than when imposing the constraint on the prior?

**(Q3)** Can MISs be used to learn more accurate yet cautious models on real medical data?

**Data sets.** To answer these research questions, we consider three types of data sets - data sampled from BNs, benchmark data sets and medical study data sets. We used three standard BNs - Asia, Cancer, and LUCAS - to generate the first five data sets. These BNs represent well-defined causal relationships between variables providing a controlled environment where domain knowledge is guaranteed to be correct. We used five data sets from UCI Machine Learning repository as benchmark data sets, namely, Haberman's Survival, Pima Indians Diabetes, Breast Cancer, Thyroid Disease, and Heart Disease. We use the same pre-processing and domain knowledge as in prior work [Yang and Natarajan, 2013] for these data sets. Finally, we used data sets from four medical studies, namely, Alzheimer's Disease Neuroimaging Initiative (ADNI), Rare diseases Survey (Rare, MacLeod et al. [2016]), Post-Partum Depression Survey (PPD, Natarajan et al. [2017]), and Nulliparous Pregnancy Outcomes Study: Monitoring Mothers-to-Be (nuMoM2b, Haas et al. [2015]). The target variables ($Y$) in all the data sets are Boolean and the parents ($\boldsymbol{X}$) are ordinal variables. Table 4 details the size of the datasets, the Boolean target variables considered in our experiments, and the parent variables of the target together with the kind of monotonic influence they have on the target.

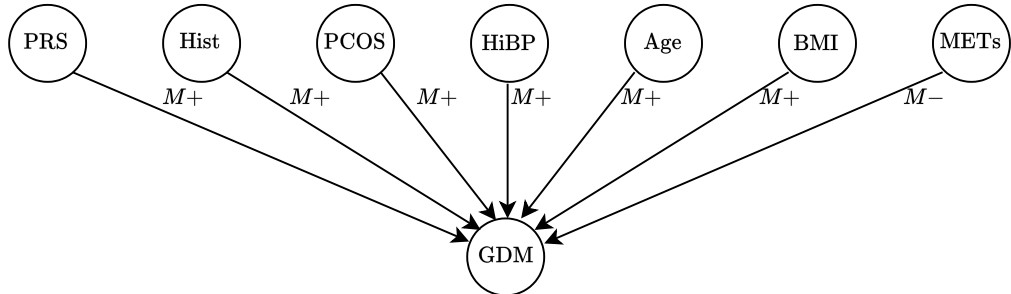

Figure 1: CN structure for the nuMoM2b domain. The risk of Gestational Diabetes Mellitus (GDM) is influenced by seven risk factors – the genetic predisposition to Diabetes as measured by a Polygenic Risk Score (PRS), family history of Diabetes (Hist), the presence of Polycystic Ovary Syndrome (PCOS), the presence of high Blood Pressure at start of pregnancy (HiBP), the age at start of pregnancy $\geq 35$ (Age), the Body Mass Index at start of pregnancy $\geq 25$ (BMI) and the amount of physical activity measured in Metabolic /Equivalents of Time $\geq 450$ (METs). All risk factors except METs positively monotonically influence the risk of GDM, while METs negatively monotonically influences the risk of GDM.

| Data set | BN | CN-IDM | | CN-IDM-MIS-P | | CN-IDM-MIS | |
|---|---|---|---|---|---|---|---|
| | | Accuract | Uncertainty | Accuracy | Uncertainty | Accuracy | Uncertainty |
| Asia | 0.854±0.02 | **0.855±0.02** | 0.070±0.04 | 0.853±0.02 | 0.053±0.02 | 0.853±0.02 | **0.016±0.02** |
| Cancer | 0.986±0.00 | **0.988±0.00** | 0.037±0.01 | **0.988±0.00** | 0.037±0.01 | **0.988±0.00** | **0.027±0.02** |
| LUCAS-a | 0.772±0.03 | **0.852±0.03** | 0.249±0.11 | **0.852±0.03** | 0.228±0.15 | 0.845±0.02 | **0.197±0.11** |
| LUCAS-b | 0.828±0.01 | 0.826±0.01 | 0.027±0.02 | 0.826±0.01 | 0.027±0.02 | **0.829±0.01** | **0.008±0.01** |
| LUCAS-c | 0.820±0.01 | **0.840±0.02** | 0.114±0.09 | 0.839±0.02 | **0.096±0.10** | 0.839±0.02 | **0.096±0.10** |
| Haberman | **0.745±0.02** | 0.675±0.18 | 0.381±0.34 | 0.712±0.17 | 0.289±0.27 | 0.703±0.16 | **0.201±0.29** |
| Diabetes | 0.679±0.03 | **0.799±0.05** | 0.462±0.05 | 0.758±0.06 | 0.288±0.04 | 0.730±0.05 | **0.122±0.03** |
| Thyroid | 0.941±0.04 | 0.943±0.04 | 0.059±0.06 | **0.945±0.04** | **0.011±0.02** | **0.945±0.04** | **0.011±0.02** |
| Heart Disease | 0.562±0.04 | **0.635±0.08** | 0.482±0.06 | 0.629±0.10 | 0.303±0.05 | 0.634±0.11 | **0.192±0.11** |
| Breast Cancer | 0.668±0.09 | 0.758±0.11 | 0.394±0.19 | **0.759±0.07** | 0.268±0.19 | 0.739±0.11 | **0.104±0.08** |
| ADNI | 0.812±0.06 | **0.843±0.08** | 0.101±0.04 | 0.834±0.08 | 0.056±0.04 | 0.827±0.08 | **0.030±0.04** |
| Rare | 0.667±0.04 | 0.696±0.04 | 0.155±0.05 | 0.690±0.04 | 0.137±0.05 | **0.706±0.03** | **0.055±0.03** |
| PPD | 0.751±0.07 | **0.766±0.09** | 0.288±0.11 | 0.748±0.06 | 0.167±0.12 | 0.762±0.07 | **0.029±0.03** |
| nuMoM2b | 0.949±0.01 | **0.965±0.00** | 0.045±0.01 | 0.964±0.00 | 0.030±0.01 | 0.964±0.00 | **0.010±0.00** |

Table 2: Accuracy of precise conditional CPTs learning using the Dirichlet prior (BN), and Accuracy and Uncertainty of CCSs learned using IDM (CN-IDM), using IDM with monotonicity constraints on the prior (CN-IDM-MIS-P) and using IDM with monotonicity constraints on the posterior (CN-IDM-MIS) for each data set.

**Methods.** We compare our algorithm (discussed in the previous section and denoted here as CN-IDM-MIS) against two baselines – (i) a CN estimator based on the pure IDM (denoted as CN-IDM); (ii) a CN estimator that applies constraints in the Imprecise Dirichlet Prior (denoted as CN-IDM-MIS-P, de Campos and Cozman [2005]). To illustrate the difference in the types of data sets, we also present the results for a precise BN estimator with a Dirichlet prior (denoted as BN).

We set the ESS $s = 2$ for all data sets and models. Additionally, we set $\epsilon = 0.01$ for all the constraints in the BN data sets and $\epsilon = 0.001$ for the other data sets. The Python code used for the experiments is freely available in a public repository[1].

[1] https://github.com/saurabhmathur96/credal-cpd

We perform inference in the CN models by interval-dominance. If neither value of the Boolean target interval-dominates the other, we mark that data point as uncertain and do not make an inference for it. For BNs we perform inference by thresholding the positive probability at $\geq 0.5$.

**Metrics.** We evaluate the methods using three metrics – the fraction of uncertain data points (*uncertainty*), *accuracy* over non-uncertain data points, and a utility-discounted accuracy (*discounted accuracy*).

Compared to its Bayesian counterpart, whose uncertainty is zero by construction, a credal method typically achieves higher accuracy at the price of a growing uncertainty (see, e.g., Antonucci and Corani [2017]). The discounted accuracy provides a summary of such a trade-off: this perfor-

| Data set | BN | CN-IDM | CN-IDM-MIS-P | CN-IDM-MIS |
|---|---|---|---|---|
| Asia | **0.854±0.02** | 0.830±0.02 | 0.834±0.02 | 0.847±0.02 |
| Cancer | **0.986±0.00** | 0.970±0.01 | 0.970±0.01 | 0.974±0.01 |
| LUCAS-a | 0.772±0.03 | 0.762±0.02 | 0.768±0.03 | **0.776±0.02** |
| LUCAS-b | **0.828±0.01** | 0.818±0.01 | 0.818±0.01 | 0.826±0.01 |
| LUCAS-c | **0.820±0.01** | 0.801±0.03 | 0.805±0.03 | 0.805±0.03 |
| Haberman | **0.745±0.02** | 0.653±0.14 | 0.685±0.13 | 0.700±0.13 |
| Diabetes | 0.679±0.03 | 0.659±0.02 | 0.682±0.04 | **0.702±0.04** |
| Thyroid | **0.941±0.04** | 0.916±0.04 | 0.941±0.04 | 0.941±0.04 |
| Heart Disease | 0.562±0.04 | 0.571±0.04 | 0.589±0.07 | **0.601±0.07** |
| Breast Cancer | 0.668±0.09 | 0.655±0.08 | 0.689±0.07 | **0.713±0.09** |
| ADNI | 0.812±0.06 | 0.806±0.06 | 0.814±0.07 | **0.815±0.07** |
| Rare | 0.667±0.04 | 0.665±0.03 | 0.663±0.03 | **0.694±0.03** |
| PPD | 0.751±0.07 | 0.682±0.05 | 0.702±0.04 | **0.754±0.06** |
| nuMoM2b | 0.949±0.01 | 0.944±0.01 | 0.950±0.00 | **0.959±0.00** |

Table 3: Discounted accuracy of precise conditional CPTs learning using the Dirichlet prior (BN), and the conditional credal sets learned using IDM (CN-IDM), using IDM with monotonicity constraints on the prior (CN-IDM-MIS-P) and using IDM with monotonicity constraints on the posterior (CN-IDM-MIS) for each data set.

| Data set | $|\mathcal{D}|$ | $Y$ | $X$ |
|---|---|---|---|
| Asia | 2000 | dysp | bronc$^+$, either$^+$ |
| Cancer | 2000 | Cancer | Pollution$^+$, Smoker$^+$ |
| LUCAS-a | 2000 | Smoking | Peer_Pressure$^+$, Anxiety$^+$ |
| LUCAS-b | 2000 | Lung_cancer | Smoking$^+$, Genetics$^+$ |
| LUCAS-c | 2000 | Coughing | Allergy$^+$, Lung_cancer$^+$ |
| Haberman | 306 | survive | nodes$^-$, year$^+$, age$^-$ |
| Diabetes | 392 | Outcome | Age$^+$, Pregnancies$^+$, BMI$^+$, DiabetesPedigreeFunction$^+$ |
| Thyroid | 185 | Hyperthyroid | T3_resin$^+$, T3$^+$, TSH$^+$, TSH_diff$^+$, T4$^+$ |
| Heart Disease | 297 | heart_disease | sex_male$^+$, age$^+$, trestbps$^+$, chol$^+$, diabetes$^+$ |
| Breast Cancer | 277 | recurrence | age$^+$, menopause$^+$, deg_malig$^+$, tumor_size$^+$, irradiat$^-$ |
| ADNI | 336 | DXCURREN | MMSCORE$^-$, AGE$^+$, PTGENDER$^+$ |
| Rare | 291 | rare | online_discuss$^+$, memorialize$^+$, specialists$^+$, review_hospital$^+$ |
| PPD | 173 | ppd | partner_support$^-$, life_stress$^+$, maternity_blues$^+$, unplanned$^+$ |
| nuMoM2b | 3657 | GDM | Age$^+$, BMI$^+$, PRS$^+$, Hist$^+$, PCOS$^+$, METs$^-$, HiBP$^+$ |

Table 4: The number of examples ($|\mathcal{D}|$), the target ($Y$) and feature variables ($X$) for each of the data sets used for empirical evaluation. The data sets are of three types – BN based (rows 1–5), UCI Benchmark (rows 6–10) and medical study data (rows 11–14). A feature with the superscript + denotes a positive monotonic influence, and a feature with the superscript - denotes a negative monotonic influence.

mance descriptor accounts for uncertain data points by assigning them a score of 0.5 while scoring the correct and incorrect classifications identically to accuracy (as 1 and 0 respectively). Discounted accuracy coincides with the accuracy for models that always make predictions (like BNs).

We compute these metrics by five-fold cross-validation. Additionally, to simulate small data settings, we limit the training set size in the BN data sets to 50 data points.

**Results.**

(Q1) Rows 1–5 and 6–10 of Table 2 present the accuracy and the number of uncertain examples for the BN based and the UCI benchmark data sets. To simulate small data setting, we fixed the training set size for BN data sets to 50. The UCI benchmark data sets are both noisy and small. This can be seen by the reduction in accuracy of the precise BN estimator from the BN data sets to the UCI data sets (from 85.2% to 75.9%).

The IDM method achieves high accuracies for these data sets, but the price is being uncertain about a large number of test examples. The uncertainty rate is 10% for the BN data sets on average. This increases to 35.5% for the UCI benchmark data sets.

The methods using qualitative knowledge (CN-IDM-MIS-P and CN-IDM-MIS) reduce the number of uncertain examples relative to CN-IDM by 20.2% on average with an average relative decrease in accuracy of just 0.4% . Hence, Q1 is answered affirmatively.

(Q2) Recall that the CN-IDM-MIS-P method imposes the monotonic constraints on the imprecise Dirichlet prior while the CN-IDM-MIS method imposes the constraints directly on the posterior distribution. Rows 1–5 and 6–10 of Table 3 present the discounted accuracy for the BN based and the UCI benchmark data sets. CN-IDM-MIS achieves same or better discounted accuracy for all the data sets (same for LUCAS-c and Thyroid). The average improvement in discounted accuracy in CN-IDM-MIS relative to CN-IDM-MIS-P is 1.5%. Hence, Q2 is answered affirmatively.

(Q3) Rows 11–14 of Table 2 present the accuracy and the number of uncertain examples for the four medical data sets. The methods using qualitative knowledge reduce the uncertainty by 54.3% on average with an average reduction of 0.7% in the accuracy. Rows 11–14 of Table 3 present the discounted accuracies for the medical data sets. On average, the methods using qualitative knowledge achieve a 2.7% improvement in the discounted accuracy, and CN-IDM-MIS achieves an improvement of 3.3% in discounted accuracy over CNN-IDM-MIS. Hence, Q3 is answered affirmatively.

Table 2 shows that the data-driven credal approach CN-IDM typically achieves the highest accuracy at the price of a high uncertainty. On the other hand, precise models like BNs lack a way to abstain from prediction and as a result always make a prediction, even if the prediction might be unreliable. In such a situation, our approach CN-IDM-MIS might represent a reasonable balance. This can also be seen in the discounted accuracy scores in Table 3. Utility-discounted accuracy assigns a score of 0.5 to uncertain data points and tends to over-penalize credal models in comparison to precise models, making such a comparison unfair to the credal model [Zaffalon et al., 2012]. In this light, the fact that our approach outperforms the precise BN on discounted accuracy on many data sets indicates significant benefit from the use of qualitative knowledge.

## 5 CONCLUSION AND FUTURE WORK

It is clear that in many domains such as healthcare an interval probability estimate would suffice rather than computing a clear point estimate. For instance, if the intervals between an event occurring and not occurring do not overlap, it can result in an actionable outcome. Hence, we considered the problem of learning credal networks from data and domain-specific qualitative knowledge. We presented an IDM-based procedure to learn credal networks from data in a way that is also consistent with the qualitative knowledge expressed by monotonic influence statements. This is achieved by an iterative procedure shrinking the IDM bounds. Our empirical evaluation demonstrates that the proposed algorithm yields conditional credal sets that have higher coverage without losing much accuracy.

There are several directions for future research. First is to extend the proposed method to support other qualitative influence statements like synergies where one specifies the effect of more than one random variable on a target (for example, higher BMI with a lower HDL level increases the risk of heart attack). Next, one could consider a more general setup where the qualitative influence statements are not restricted to parent-child relations but are instead over joint distributions. Also, one could employ the recent generative AI models to provide weak knowledge. Finally, learning from multiple experts while assessing the credibility of each expert could open up human-allied learning to very large problems such as healthcare.

## ACKNOWLEDGEMENTS

The authors acknowledge the support by AFOSR award FA9550-23-1-0239, ARO award W911NF2010224, and NIH grant R01HD101246.

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
