# OpenReview forum: "Knowledge Intensive Learning of Credal Networks"
_auai.org/UAI/2024/Conference — UAI 2024 poster_

### Official Review · Reviewer_g4pN · 2024-03-19

**Q2-1 Originality-Novelty:** 3
**Q2-2 Correctness-Technical Quality:** 3
**Q2-5 Clarity Of Writing:** 3

**Q1 Summary And Contributions:**

This paper introduces a method for learning parameters of credal networks using additional domain knowledge in the form of qualitative constraints. This approach addresses a clear limitation of credal networks where intervals can become too large and where some combinations of probability distributions are not plausible. The parameters are being learned using a gradient ascent procedure. The method is compared to several baselines: precise BNs, using an imprecise Dirichlet prior, and an approach where the qualitative constraints are applied on the prior, rather than the posterior. It is shown that in terms of classification incorporating qualitative constraints can significantly reduce the uncertainty in predictions, while maintaining a good (but not best) accuracy.

**Q2-3 Extent To Which Claims Are Supported By Evidence:**

3: Good: the main claims are supported by convincing evidence (in the form of adequate experimental evaluation, proofs, (pseudo-)code, references, assumptions).

**Q2-4 Reproducibility:**

4: Excellent: key resources (e.g. proofs, code, data) are available and key details (e.g. proof sketches, experimental setup) are comprehensively described for competent researchers to confidently and easily reproduce the main results.

**Q3 Main Strengths:**

The paper introduces a novel approach for incorporating domain knowledge in the form of monotonic influence statements in credal networks, which could make credal models more applicable in practice. The paper is well-presented and clear. I believe it is a relevant and appropriate contribution to UAI.

**Q4 Main Weakness:**

The results show a clear benefit of the proposed approach compared to other methods for credal networks (CN-IDM/CN-IDM-MIS-P), mainly by  reducing the number of uncertain classifications. Compared to precise BNs, there are no obvious improvements. While the paper mentions that the chosen metric (discounted accuracy score) overpenalizes credal models, it leaves open the question of what the value is of the proposed method.

Another obvious baseline that has been excluded would be precise Bayesian networks that employ the same qualitative constraints. See e.g.
Masegosa, A. R., Feelders, A. J., & Van Der Gaag, L. C. (2016). Learning from incomplete data in Bayesian networks with qualitative influences.
International Journal of Approximate Reasoning, 69, 18-34.

**Q5 Detailed Comments To The Authors:**

There are some minor typos that I came across:
- Sometimes "Credal" is written instead of "credal". Dirichlet should b capitalized.
- Page 2: "extreme points CS" => "extreme points OF CS"
- Under Figure 1: "GestationaL diabetes" (L is missing)

**Q9 Complying With Reviewing Instructions:**

Yes

---

> ### Author Rebuttal · Authors · 2024-04-05
>
> $\textbf{Discounted accuracy.}$ One of the key benefits of credal networks is the option to abstain from making a prediction. The relative value assigned to abstaining from making a prediction is domain-dependent.
>
> To ensure consistency across data sets, we chose to evaluate the models using discounted accuracy, which assigns a score of 0.5 to the cases where the model abstains from making a prediction.  However, in high-stakes domains like healthcare, wrong decisions might be fatal. In such domains, abstaining might be assigned a higher score.
>
> Note that despite discounted accuracy over-penalizing credal appraoches, our approach outperforms the baselines for noisy and uncertain data sets (Table 5).
>
>
> $\textbf{Masegosa et al. (2016) as a baseline}$. We thank the reviewer for pointing us to this work, which is definitely a relevant reference. Yet, it should be noted that the paper focuses on learning from missing data. A comparison against our procedure would be possible by interpreting the s parameter of the IDM as an equal number of missing (virtual) instances. Yet, the method from de Campos and Cozman considered in our experiments directly works with the IDM setup and might represent a better baseline for comparisons. Notably, the method of de Campos and Cozman can be also regarded as a credal version of the pure Bayesian approach suggested by reviewer Ud9p

---

### Official Review · Reviewer_AqFD · 2024-03-19

**Q2-1 Originality-Novelty:** 3
**Q2-2 Correctness-Technical Quality:** 3
**Q2-5 Clarity Of Writing:** 3

**Q10 Ethical Concerns:**

No.

**Q1 Summary And Contributions:**

The paper develops a new iterative method for learning credal networks from data along with additional knowledge, such as knowledge about monotone relationships between variables. The method is tested against on a variety of datasets.

**Q2-3 Extent To Which Claims Are Supported By Evidence:**

4: Excellent: all claims are supported by very convincing evidence (in the form of comprehensive experimental evaluation, rigorous mathematical proofs, detailed (pseudo-)code, precise references, well-motivated and realistic assumptions) and the authors deliver what they promise.

**Q2-4 Reproducibility:**

3: Good: key resources (e.g. proofs, code, data) are available and key details (e.g. proofs, experimental setup) are sufficiently well-described for competent researchers to confidently reproduce the main results.

**Q3 Main Strengths:**

The idea of using credal networks to incorporate domain specific knowledge along with data is novel. Robust learning of credal networks is computationally challenging, and starting from weak states of knowledge gives these networks a distinct advantage, however it is also a downside if the inferences from such networks remain very uncertain. The ability to further reduce uncertainty through structural statements seems very useful in this context. The paper is mostly well written.

**Q4 Main Weakness:**

Some questions about the algorithm remain, and not all the details are sufficiently explained. The description of the mathematics behind the algorithm itself was particularly unclear to me (i.e. the barrier method). There are some issues with the formatting of citations.

**Q5 Detailed Comments To The Authors:**

* on p4, I understand that eq (10) is meant to be a transformation of the problem in eq (8), through a barrier method to capture violations of the constraints, in a way that keeps the problem feasible. Some key aspects of this transformation however remain somewhat unclear. This might well be standard, but even in that case I think it would be useful for readers to have a bit more detail to fully appreciate what is going on.

  * Why is the error squared?
  * Is the main difference that an "epsilon" is introduced to allow for small violations of the constraints?
  * Is the epsilon a variable in the optimisation, or is it fixed in advance?
  * Why is it necessary to use the barrier form of the optimisation problem? Could one not simply include the epsilon directly into the constraints?

* Check brackets around citations throughout the paper (i.e. \citet vs \citep), as there are quite many places where the formatting seems wrong.

* Could imprecision be leveraged to learn a network that instead widens the bounds to incorporate structural constraints (i.e. by disposing of data which leads to constraint violations, or some other method)?

* Can the method be extended to other types of structural knowledge about the network, beyond monotonicity?

**Q9 Complying With Reviewing Instructions:**

Yes

---

### Official Review · Reviewer_Ud9p · 2024-03-20

**Q2-1 Originality-Novelty:** 2
**Q2-2 Correctness-Technical Quality:** 3
**Q2-5 Clarity Of Writing:** 4

**Q10 Ethical Concerns:**

No.

**Q1 Summary And Contributions:**

This paper proposes an approach to learn credal nets – a generalization of Bayesian networks with credal sets for probability parameters instead of point probabilities – by augmenting data with qualitative knowledge in the form of monotonic influence statements such as “higher values of parent configuration X result in higher values of state of child Y”. The approach centers around solving the optimization problem in equation (10). Some experiments are conducted to illustrate the overall benefits of the approach.

**Q2-3 Extent To Which Claims Are Supported By Evidence:**

3: Good: the main claims are supported by convincing evidence (in the form of adequate experimental evaluation, proofs, (pseudo-)code, references, assumptions).

**Q2-4 Reproducibility:**

4: Excellent: key resources (e.g. proofs, code, data) are available and key details (e.g. proof sketches, experimental setup) are comprehensively described for competent researchers to confidently and easily reproduce the main results.

**Q3 Main Strengths:**

This work follows a long line of research on leveraging qualitative knowledge for learning probabilistic graphical models. As the authors state, there is a lot of value in such approaches for practical problems. The paper might be seen as somewhat old-fashioned in the age of generative AI, but I think the work is still useful and applicable.

Another strength of the work, which some may see as a weakness, is the simplicity of the approach. The main contribution is essentially the formulation and using a trick for solving a challenging optimization problem, as shown in equation (10).

**Q4 Main Weakness:**

As much as I enjoy seeing this sort of work, I find the scope of the work to be considerably narrow. I believe the contributions would be more substantial if other forms of knowledge would also have been incorporated in the approach. There are some interesting ideas in the last paragraph of the paper; I think incorporating some of these would make for a stronger paper.

I do not fully understand the contributions of this work beyond some papers mentioned in Section 2. For instance, I understand that this may be novel for credal nets, but I can not glean how much extra work is needed to make this work for credal nets.

I am also not sure that sufficient baselines are explored for the experiments. For instance, a potential baseline is to take a Bayesian approach to learning a Bayes net, including incorporation of the constraints from the qualitative knowledge, and then use a hyper-parameter to obtain intervals around the mode estimates for parameters. Perhaps this is too much work for a baseline; perhaps the authors can specify why they think their baselines are sufficient.

**Q5 Detailed Comments To The Authors:**

Here are some additional questions and comments:

I strongly recommend changing the title – it is currently too broad in scope. This paper does not span knowledge intensive learning for credal nets in general. Please be more descriptive about the sort of knowledge leveraged in the title.

There are citation issues throughout the paper; see the first paragraph for example.

I recommend adding more references on credal nets and on knowledge augmentation in probabilistic graphical models. Some references are mentioned in Section 2 but I believe there are many more that would be relevant. As the authors note, there is a substantial amount of prior work in this area for Bayesian models as well as other graphical models. It is also important to explain the additional novelty beyond prior work.

Why is capital C used for Credal in the main paper? I understand why B is capitalized for Bayesian of course. On a related note, I recommend using capital letter for words such as “Bayesian” in all references.

It may be clarifying to mention L(q) in equation (8) since it is referred to later, after equation (10).

A longer discussion about the evaluation metrics and why the “uncertainty” metric is important would be useful.

**Q9 Complying With Reviewing Instructions:**

Yes

---

> ### Author Rebuttal · Authors · 2024-04-05
>
> $\textbf{Other forms of knowledge.}$ The proposed framework is sufficiently general to accommodate any form of knowledge that can be encoded as linear inequality constraints. While we present empirical results using monotonic influences, synergistic influences can also be expressed as linear constraints on the conditional distribution in a very similar manner. (See Yang and Natarajan ECML-PKDD (2013)).
>
> The framework can also accommodate knowledge that can be expressed as linear equality constraints such as context-specific independence. Consider a variable $Y$ having three parents $X_1, X_2, X_3.$ The context-specific independence that  $Y\perp\mkern-10mu\perp X_2\mid X_3 =1$ is equivalent to the constraint
> $$P(Y \mid X_1, X_2=x_2, X_3=1) = P(Y | X_1, X_2=x_2, X_3=1) \forall x_2  \in \text{Domain}(X_2)$$
> and can be encoded by adding a penalty term similar to the one in equation (9).
>
> $\textbf{Baseline}$ Credal networks are a generalization of BNs. So, the baseline CN-IDM-MIS-P (de Campos and Cozman (2005)) can be seen as a generalization of the pure Bayesian approach. It constructs an imprecise dirichlet prior (which is a generalization of the dirichlet prior) that encodes the constraints, computes the posterior, and uses the lower and upper bounds of the posterior predictive.

---

### Official Review · Reviewer_24J8 · 2024-03-22

**Q2-1 Originality-Novelty:** 3
**Q2-2 Correctness-Technical Quality:** 2
**Q2-5 Clarity Of Writing:** 2

**Q1 Summary And Contributions:**

In this paper the authors propose a new way of learning credal networks, which are imprecise-probabilistic variants of Bayesian networks.
Their proposed way of learning combines data with qualitative assessments.

The authors start from the well-known Imprecise Dirichlet Model (IDM), and note that this may yield very wide (imprecise) sets of probabilities, which reduce the capabilities to make decisions when the data is scarce.
They solve this by adjusting the IDM using Monotonic Influence Statements (MISs).
The main idea is to correct the credal set obtained by the IDM by taking the conditions of the MISs into account.
As such, a smaller credal set is obtained, which reflects both the data through the IDM as well as qualitative assessments encoded by the MISs.
Because the obtained credal set is smaller, this reduces the indecisiveness, which they expirementally validate.

**Q2-3 Extent To Which Claims Are Supported By Evidence:**

2: Fair: the main claims are somewhat supported by evidence (but the experimental evaluation may be weak, or does not match entirely with the claims, important baselines may be missing, proofs contain important ideas but lack rigor, algorithmic details are only discussed superficially, references are imprecise, assumptions are not sufficiently motivated or explicated, etc.).

**Q2-4 Reproducibility:**

3: Good: key resources (e.g. proofs, code, data) are available and key details (e.g. proofs, experimental setup) are sufficiently well-described for competent researchers to confidently reproduce the main results.

**Q3 Main Strengths:**

The idea of obtaining a credal set by combining the IDM and qualitative assessments, is very valuable to my mind.
I agree with the main reasoning of the authors, and also with their conclusion that this is a natural way of learning that leads to less indecisiveness. This idea come natural in the context of credal networks.

The paper introduces the idea by Table 1, which is a good way to do so: it is a simple example that still shows the idea. I also appreciate the pseudocode of their algorithm.

Moreover, the paper has a good experimental section. The conclusions drawn from it, seem indeed to validate sufficiently the claims.

I also appreciate that the authors made their code available through Anonymous GitHub, which is both clear and guarantees anonymity.

**Q4 Main Weakness:**

In my opinion, the paper lacks from a theoretical point of view.

As I indicated above, I like the main idea of combining the IDM with qualitative assessments, and I think that the proposal of the paper is valuable.
Table 1 is used to informally explain the main idea.
The left half shows the credal set obtained through IDM.
If I understand it correctly, the right half displays corrections taking the qualitative assessments by MIS into account, by making lower bounds larger, and making upper bounds smaller.
However, the paper does not talk about the properties of the obtained credal set.
Clearly, it is not coherent (see for instance the last line of the table where $P(G=0\vert 1,1)\leq0.2$ but $P(G=1\vert 1,1)\geq0.3$ which are two bounds that don't coincide) but it can be corrected to a consistent one (by, for instance, taking the intersection – would that operation make sense in this case?).
But more importantly, what properties does this operation yield? Always closed and convex? It seems to me that this is not the case, and that the resulting credal set may even be empty.
This latter case would seem rather detrimental, but can perhaps be avoided by considering a larger hyper parameter $s$ of the IDM.
I think much more discussion about this is needed.

Moreover, in Section 3.1, the authors seem to solve a related but rather different problem.
In Eq. (8) they find the resulting credal set by maximizing a sum.
Why do they want to maximize a sum? Why don't they care about the lower probabilities, in Eq. (7)?
I doubt that this optimization reduces to the original problem (I don't think so, since the optimization never leads to an empty credal set), but the paper does not talk about this.

As a conclusions, I think that the paper contains a valuable idea, but that it lacks in the discussion of the theoretical properties.

**Q5 Detailed Comments To The Authors:**

Here are my detailed comments:

Page 1, right column, last line of second paragraph: I think that the authors intend to say that these approaches can easily result in MORE imprecise estimates (as CNs generally result in imprecise estimates, even for large data sets, but perhaps only slightly imprecise)

Page 2, Section 2, paragraph starting with "Credal networks": I think that the IDM could be explained more clearly and concisely, by for instance making explicit from the outset that a credal set is a non-empty, convex and closed set (and not just any set). Without this, the extreme points mentioned in the subsequent section are not guaranteed to exist. In the current version, the authors mention only in the following paragraph what properties a credal set has.

Page 4, Table 1: I think the first line has wrong labels for A and B (or perhaps another line is wrong). Also, I fail to derive some corrections myself; an indication on how to do this would be welcome. In the caption text, the dot after (B) is redundant.

Page 4, left column, first paragraph: Eq. 8 should read Eq. (8).

Page 4, Eq. (11): there is a $\forall$ missing.

Page 5, Algorithm 2: On the second line of Input, $\mathrm{Dom}((\mathbf{X})$ should be $\mathrm{Dom}(\mathbf{X})$ (it has a redundant opening bracket). On line 2 of Output, I would initialize t=1, as now t has no initialization.

Page 6, Figure 1: In the caption, "presense" should read "presence".

Page 6, Table 2: For consistency, I think the first "Acc" should read "Accuracy".

Pages 7 and 8: Rows 5 -- 10 should read 6 -- 10, and 10 -- 14 should read 11 -- 14, I believe.

**Q9 Complying With Reviewing Instructions:**

Yes

---

> ### Author Rebuttal · Authors · 2024-04-05
>
> $\textbf{Coherence of intervals}$ As the reviewer has rightly pointed out, our intervals, as shown by the example in Table 1, are not always reachable. This is because we perform two independent optimizations for the lower and upper bounds (Alg 1, lines 2 and 3). However, by construction, these constraints define a non-empty credal set. Specifically, if there exists a value of $q$ between the given bounds that satisfies the monotonicity constraints, then the barrier penalty method is guaranteed to converge to that value.
> So, we can shrink the intervals after the optimization to make them reachable (e.g., “Probability Intervals: A tool for Uncertain Reasoning” by de Campos, Huete and Moral, 1994).
>
> $\textbf{Maximization}.$ Since the description of the maximization and minimization procedures are symmetric, we explain our approach in terms of maximization. However, our approach involves performing a minimization and a maximization to compute the lower and upper bound respectively (The last sentence of section 3.1 says "We analogously proceed for the minimization task.")
>
> Additionally, the maximization in equation (8) involving a sum of the form $\sum q_{y\mid x}$ is equivalent to maximizing each individual $q_{y\mid x}$ in (7) because they are separate, non-negative values.

---

### Official Review · Reviewer_JgZS · 2024-03-27

**Q2-1 Originality-Novelty:** 3
**Q2-2 Correctness-Technical Quality:** 3
**Q2-5 Clarity Of Writing:** 4

**Q1 Summary And Contributions:**

The authors consider  the problem of learning credal networks from data and domain-specific qualitative knowledge. The empirical evaluation demonstrates that the proposed algorithm yields conditional credal sets that have higher coverage without losing much accuracy.

**Q2-3 Extent To Which Claims Are Supported By Evidence:**

3: Good: the main claims are supported by convincing evidence (in the form of adequate experimental evaluation, proofs, (pseudo-)code, references, assumptions).

**Q2-4 Reproducibility:**

3: Good: key resources (e.g. proofs, code, data) are available and key details (e.g. proofs, experimental setup) are sufficiently well-described for competent researchers to confidently reproduce the main results.

**Q3 Main Strengths:**

Strong experimental results

**Q4 Main Weakness:**

Some sentences become not clear when using a lot of acronyms

**Q5 Detailed Comments To The Authors:**

This paper presents a well-written and compelling study that significantly contributes to the field. The authors have done an excellent job of clearly articulating their research objectives and methodology, making it easy for readers to follow their approach. The results presented are both robust and convincing. Additionally, the discussion of the findings effectively highlights their significance and implications for future research in the field.

I would like to have information about learning times.

**Q9 Complying With Reviewing Instructions:**

Yes

---

### Meta-Review · Area_Chair_gNnP · 2024-04-17

All reviewers generally believe there is an interesting contribution that is relevant to UAI, and generally that the way of asserting background knowledge is appropriate for credal networks.

At least two reviewers did not believe that the general claim made by the title was justified by the paper.  A more specific title might bring a better audience to the paper.